# Fermi level-tuned optics of graphene for attocoulomb-scale quantification of electron transfer at single gold nanoparticles

Qing Xia[1,2], Zixuan Chen [1,2], Pengwei Xiao[1], Minxuan Wang[1], Xueqin Chen[1], Jian-Rong Zhang[1], Hong-Yuan Chen[1] & Jun-Jie Zhu [1]

Measurement of electron transfer at single-molecule level is normally restricted by the detection limit of faraday current, currently in a picoampere to nanoampere range. Here we demonstrate a unique graphene-based electrochemical microscopy technique to make an advance in the detection limit. The optical signal of electron transfer arises from the Fermi level-tuned Rayleigh scattering of graphene, which is further enhanced by immobilized gold nanostars. Owing to the specific response to surface charged carriers, graphene-based electrochemical microscopy enables an attoampere-scale detection limit of faraday current at multiple individual gold nanoelectrodes simultaneously. Using the graphene-based electrochemical microscopy, we show the capability to quantitatively measure the attocoulomb-scale electron transfer in cytochrome c adsorbed at a single nanoelectrode. We anticipate the graphene-based electrochemical microscopy to be a potential electrochemical tool for in situ study of biological electron transfer process in organelles, for example the mitochondrial electron transfer, in consideration of the anti-interference ability to chemicals and organisms.

---

[1] State Key Laboratory of Analytical Chemistry for Life Science, School of Chemistry and Chemical Engineering, Nanjing University, 163 Xianlin Ave, 210023 Nanjing, China. [2] These authors contributed equally: Qing Xia, Zixuan Chen. Correspondence and requests for materials should be addressed to J.-J.Z. (email: jjzhu@nju.edu.cn)

Electron transfer is of great interest in many research of basic chemical and biological phenomena[1–3], molecular electronics[4,5] and energy materials[6–8]. In particular, measurement of electron transfer at single-molecule level is critical to the in situ study of biological processes[9–11]. Electrochemical detection methods provide a powerful analytical tool for understanding and tracing local electron transfer in molecules adsorbed on the electrode, and substantial advances have been made for this aim. Common electrochemical detection strategies for local electron transfer reactions fall into two main categories: scanning electrochemical microscopy (SECM)[12–14] and plasmonic-based electrochemical current imaging (P-ECi)[15–18]. SECM measures the local current by scanning a microelectrode across the surface, and found abundant applications. P-ECi offers a faster image rate and an extremely sensitivity to the refractive index change of chemical species between oxidized and reduced states. The above electrochemical technologies show the potential to be used to measure local electron transfer reactions at single-molecule level. However, one of the challenges involved in this aim is the detection limit. The key issue is the signal-to-noise ratio determined by the background current in circuit or interference from chemical species. Many efforts have been made to resolve the current at lower ranges, such as ultramicroelectrodes[19,20] and surface-enhanced Raman spectroscopy[21,22]; however, the detection limit is normally restricted in the picoampere to nanoampere range[13,15]. It is highly desirable to develop a new electrochemical detection strategy avoiding above interferences.

Graphene is an ideal two-dimensional material for developing abundant photonics and optoelectronics devices, such as displays, optical modulators, and plasmonic devices[23,24]. Furthermore, graphene has been widely used to fabricate working electrodes in a variety of electrochemical methods because of its appealing flexible, transparent properties, and low capacitance[17,25–27]. However, the intrinsic Fermi level-controlled optoelectronic properties of graphene have barely been studied for the measurement of electrochemical reactions. One of possible reasons is

the weak scattering (<0.1%) and absorption (~2.3%) of single-layer graphene[23,28], making it invisible with most conventional microscopies. The optical conductivity of graphene in the visible region can be efficiently modulated by the Fermi level and charged carrier density, involving variation of interband transitions[28–30]. Interband transitions correlate with the absorption and scattering cross section, offering a potential way to directly measure electron transfer reactions with imaging technologies.

In this work, our observation highlights the Fermi level-responsive Rayleigh scattering of graphene and attached plasmonic nanoparticles. We construct a theoretical model that converts the scattering intensity to the local current density based on experimental results. Accordingly, we develop a unique graphene-based electrochemical microscopy (GEM) technique that makes a straightforward advance in the detection limit. Contrary to the conventional optical electrochemical methods using the change in refractive index as probes, for example, P-ECi, GEM directly measure the in situ-transferred electron charges, avoiding interferences from background current noise and chemical species. In order to enhance the Rayleigh scattering and electron transfer rate, plasmonic gold nanostars (GNS) are immobilized on graphene surface and act as nanoelectrodes. Results reveal that GEM illustrates an ultralow faraday current detection limit ($4.5 \times 10^{-18}$ A) at single nanoelectrodes. Using GEM, we successfully show the potential to measure electron transfer in single cytochrome c molecules, which is an essential redox protein involved in the mitochondrial electron transfer.

## Results

**Construction of graphene-based electrochemical microscopy.** The construction of a three-electrode electrochemical cell for GEM is schematically illustrated in Fig. 1a. A 47-nm-thick gold film is deposited on a cover slide with a 4-mm-diameter hole in center. A graphene layer is transferred onto the hole and acts as the working electrode, which is immersed in electrolyte (0.1 M KNO$_3$) with

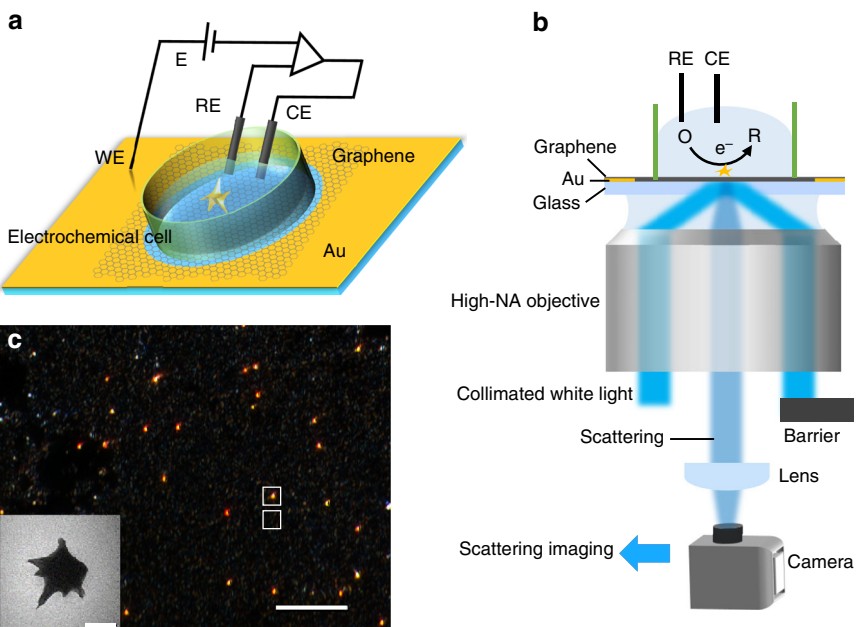

**Fig. 1** Schematic illustration of construction of the optical setup. **a** Schematic illustration of the construction of electrochemical cell, where WE, RE, and CE are working electrode, reference, and counter electrodes, respectively. A graphene layer is transferred onto a gold-coated cover slide with a 4-mm-diameter hole in center, on which attaching a 3.5-mm-diameter PDMS electrochemical cell to avoid reactions on the gold film. **b** Schematic illustration of the total internal reflection dark-field microscope. **c** Scattering image of single gold nanostars on the graphene layer. Scale bar is 10 μm. Inset is the transmission electron microscopy image of a gold nanostar. Scale bar is 50 nm

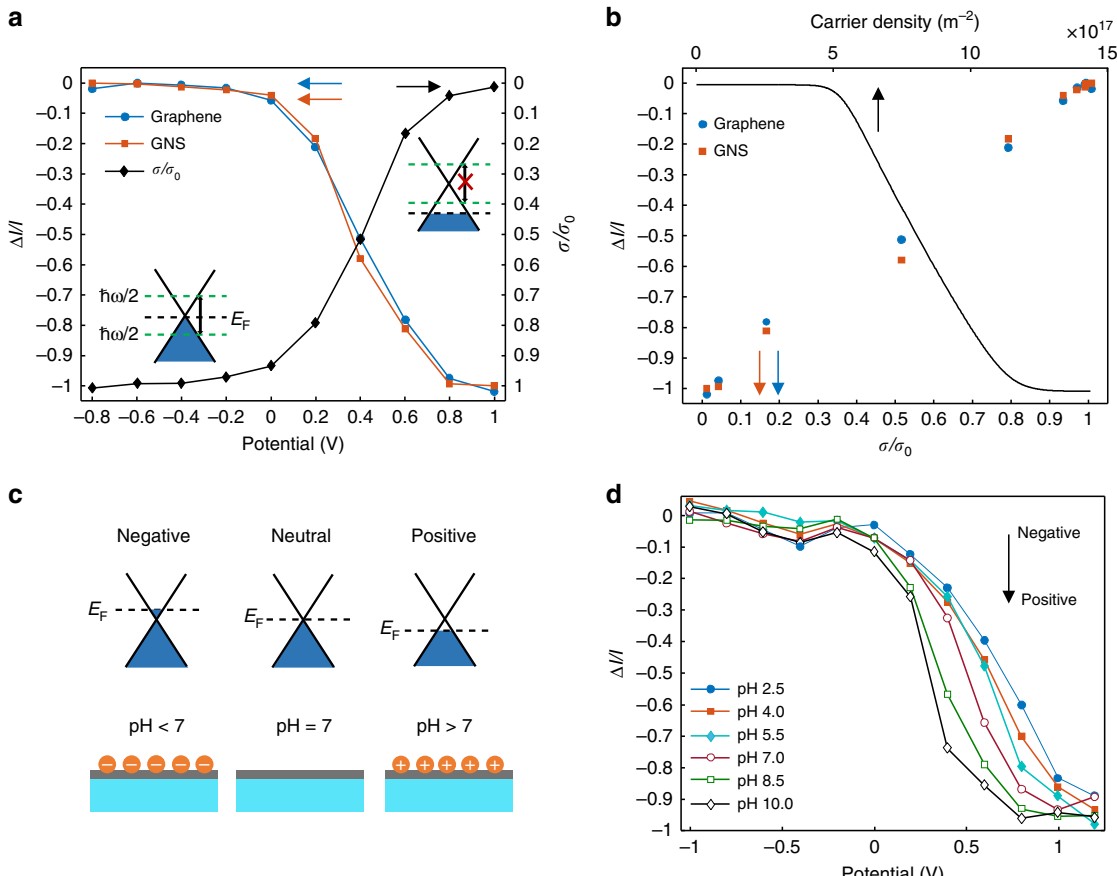

**Fig. 2** Correlation of the scattering and optical conductivity of graphene. **a** Potential dependence of relative scattering change of a gold nanostar (red), neighboring graphene (blue), and the optical conductivity of the graphene (black). Insets show the band structures of graphene at corresponding potentials. **b** Correlation of the relative scattering change and the optical conductivity (red and blue), and the theoretical correlation of graphene's carrier density and relative scattering change (black line). **c** Introduction of excess charges doping in the graphene layer with hydroxyl and hydroxonium ions. Insets show the corresponding energy diagrams. **d** Potential dependence of relative scattering change of gold nanostars with different surface condition: from negative (top) to positive (bottom) charge

a Ag/AgCl reference and a platinum counter electrode. All potentials mentioned in this work are relative to the reference. Single-gold nanostars, which demonstrate a uniform scattering over a broad range of wavelength, from visible to near infrared region (Supplementary Fig. 1), are immobilized on the graphene and act as nanoelectrodes. Scattering images are captured by our homemade total internal reflection dark-field microscope[31]. As shown in Fig. 1b, the fabricated electrochemical cell is placed above the objective, where a collimated white light from a laser-driven light source is directed onto the cover slide via the objective, and then scattered by the graphene and GNS. A barrier is placed at the back focus plane of the objective to stop the reflected light and only the scattering light is directed to a camera to form dark-field scattering images (Fig. 1c). The uniform white background reveals the Rayleigh scattering of graphene. Such weak scattering is not suitable for following experiments. In contrast, bright red scattering spots are assigned to individual GNS, contributed by the far-field incident light and the near-field scattering from underneath graphene together. According to this, GNS may have the ability to enhance the scattering of underneath graphene, and the effective enhanced area is the near-field scattering cross section of GNS[32,33].

**Correlation of surface charge and scattering intensity**. The hypothesis of plasmonic enhancement of GNS is verified by

investigating the scattering intensity of the graphene and GNS modulated by applied potentials. Figure 2a displays the relative scattering change ($\Delta I/I$) of a single GNS and neighboring graphene (Fig. 1c, open squares) at different potentials ($E$). Note that the background scattering has been deducted before for clarity. Simultaneously, we measure the optical conductivity ($\sigma$) of the graphene sample (details in "Methods" section). Notably, the strong $E$-dependent $\Delta I/I$ of GNS and the graphene are both in good agreement with the change in $\sigma$, which can be qualitatively understood from different electronic band structures[34–37]. At low potentials ($E < -0.2$ V), the Fermi level of graphene ($E_F$) is close to the Dirac point, leaving the graphene with low carrier density. The optical conductivity remains constant at a high level, and interband transitions occur when electrons are excited by incident photons ($\hbar\omega$), resulting in a strong absorption and scattering[36,37]. When $E$ is higher than $-0.2$ V, a strong $E$-dependent scattering is observed. In this range, $E_F$ gets close to the transition threshold ($E_F = 1/2\hbar\omega$) due to the positive holes accumulation, and interband transitions start to be forbidden. Higher than 0.8 V, $E_F$ is far away from the transition threshold ($E_F > 1/2\hbar\omega$), and the strongly hole-doping leaves the lowest optical conductivity, as well as the scattering intensity. We plot $\Delta I/I$ to $\sigma$ in Fig. 2b, and find a good linear relationship, $\Delta I/I = \sigma/\sigma_0 - 1$, where the quantum conductivity $\sigma_0$ is defined as $e^2/4\hbar$. The $E_F$-dependent $\Delta I/I$ consequently yields the correlation of graphene's carrier density ($n_c$) and $\Delta I/I$, since we have $E_F = \hbar v_F(\pi n_c)^{1/2}$. We calculate

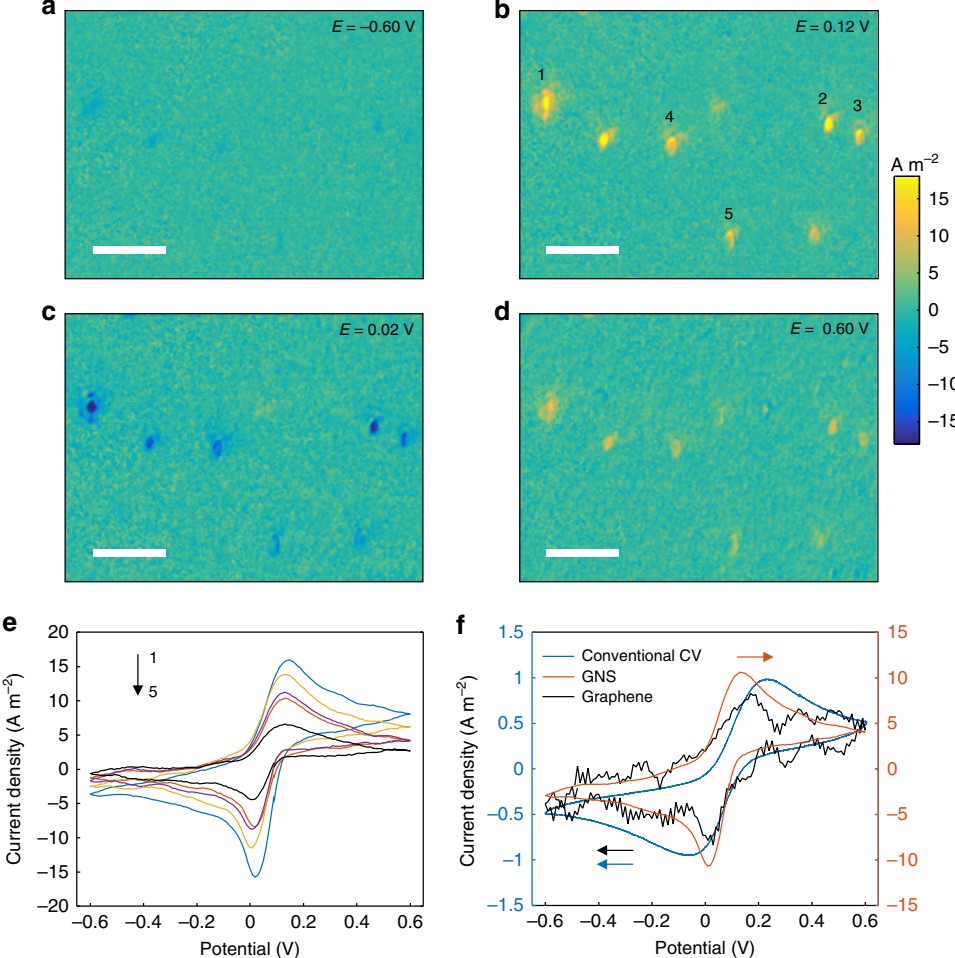

**Fig. 3** Electrochemical reaction of potassium ferricyanide at single gold nanostars. **a–d** Electrochemical current images of multiple gold nanostars on the graphene at different potentials during continuous cycling of the potential between −0.6 V and 0.6 V at a rate of 0.1 V s$^{-1}$ (see Supplementary Movie 1). Scale bar is 5 μm. The electrolyte is 0.1 M KNO$_3$ containing 1.0 mM K$_3$[Fe(CN)$_6$]. **e** Cyclic voltammograms of 5-labeled gold nanostars extracted from the electrochemical current images. **f** Averaged cyclic voltammogram of the 5 gold nanostars (red), cyclic voltammogram of graphene area (black), and the conventional cyclic voltammogram measured with a potentiostat (blue)

$n_c$-dependent $\Delta I/I$ (details in "Methods" section) and $\Delta I/I$ scales down with $n_c$ in the range from $5 \times 10^{17}$ to $1 \times 10^{18}$ m$^{-2}$ (Fig. 2b, black line),

$$n_c = A\Delta I/I, \qquad (1)$$

where $A$ is calculated to be $-6.9 \times 10^{17}$ m$^{-2}$ from the slope of fitting curve.

The correlation of $\Delta I/I$ and $n_c$ offers a direct way to measure a redox reaction taking place on graphene that involves electron transfer processes. For example, in an inner-sphere electrode reaction, such as redox reaction of Fe(CN)$_6^{3-/4-}$, there is a strong interaction of the reactant and the electrode[38,39]. It introduces excess charged carriers and hence a shift of Fermi level to the graphene[27,40]. In order to further understand the contribution of excess charges to the scattering intensity, we change the electrolyte pH from 2.5 to 10 with phosphate buffered saline (PBS) (see "Methods" section), leaving the graphene with excess negative and positive charged carriers, respectively[41]. Negative charges drive $E_F$ of graphene shift above the Dirac point, and with positive charges $E_F$ is expected to shift below the Dirac point (Fig. 2c). For comparison, we investigate $E$-dependent $\Delta I/I$ of GNS with different charges (Fig. 2d). Indeed, we found an apparent shift of the threshold potential from 0.7 V (negative charge) to 0.3 V (positive charge). That is to say, when an inner-sphere redox

reaction takes place on the graphene, both the electric double layer (EDL) charging and electron transfer processes contribute to $n_c$. However, the contribution from EDL charging is exceedingly smaller than electron transfer because of the low capacitance of graphene (Supplementary Fig. 2).

**Imaging the current density at single nanoelectrodes.** We have known the linear relationship of $n_c$ and $\Delta I/I$ according to Eq. (1). The carrier charge density in graphene is the opposite of charge density in solution[42], thus the charging current density $i_c$ is given by

$$i_c = e\frac{dn_c}{dt} = Ae\frac{d(\Delta I/I)}{dt}, \qquad (2)$$

where $e$ is the charge of a single carrier. Equation (2) reveals that the charging current density can be obtained by time derivative of $\Delta I/I$ (Supplementary Fig. 2). When an inner-sphere electron transfer reaction[38,39], such as Fe(CN)$_6^{3-/4-}$, takes place on the graphene surface, $n_c$ is contributed by two types of charges: $n_c = n_{EDL} + n_{ET}$, where $n_{EDL}$ and $n_{ET}$ is the carrier density induced by charging and electron transfer, respectively. Thus, the faraday current $i$ can be easily calculated from $\Delta I/I$ via Fick's law (details

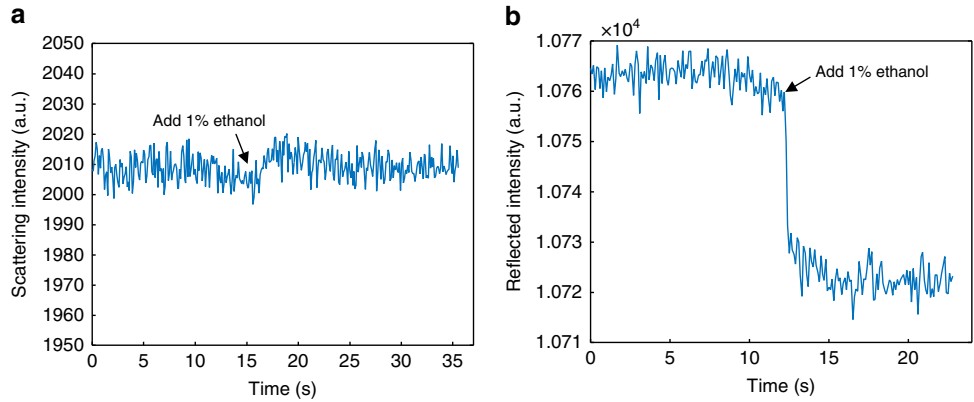

**Fig. 4** Refractive index dependence of scattering and reflected intensity. **a** Time course of the scattering intensity of a single gold nanostar. **b** Time course of the total internal reflection intensity of the same area

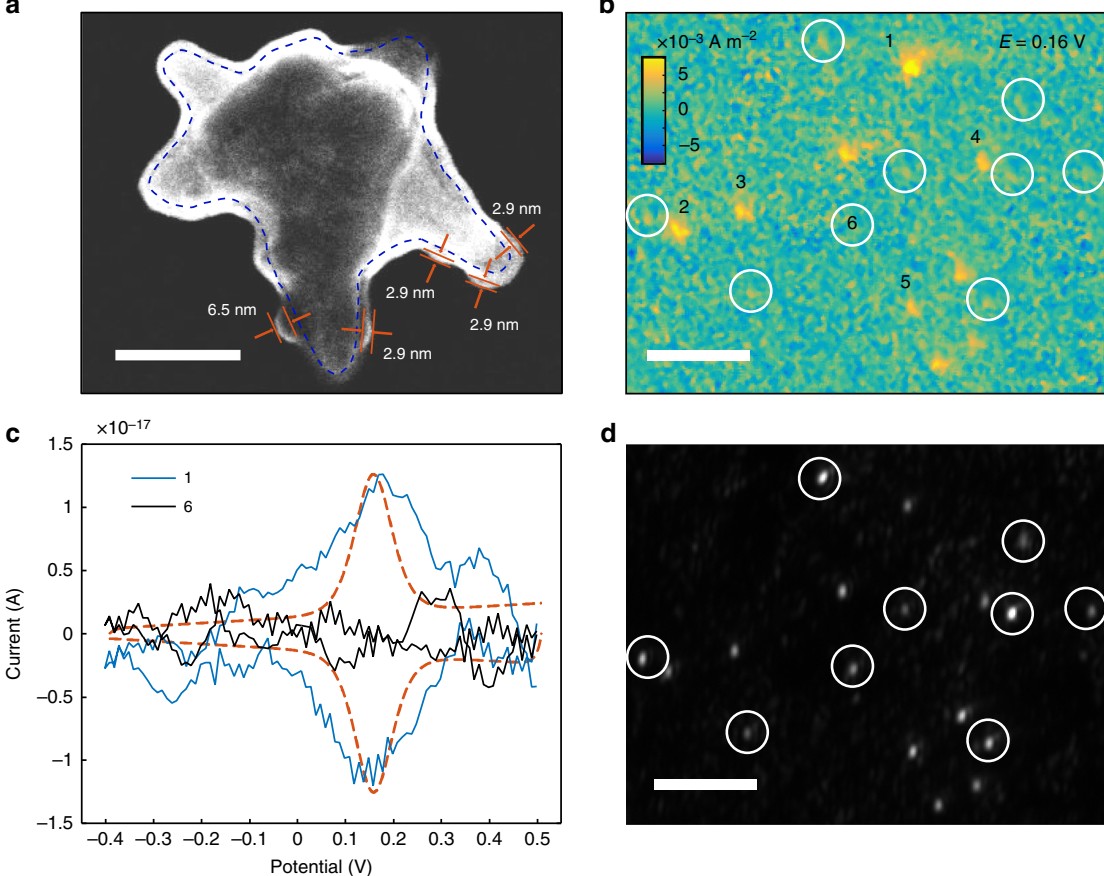

**Fig. 5** Electron transfer measurement in cytochrome c at single gold nanostars. **a** Scanning transmission electron microscopy imaging of a gold nanostar with adsorbed cytochrome c. Scale bar is 50 nm. **b** Current density imaging of multiple gold nanostars with and without (open circles) cytochrome c modification at 0.16 V during continuous cycling of the potential between −0.4 V and 0.5 V at a rate of 0.1 V s⁻¹. Scale bar is 5 μm. **c** Cyclic voltammograms of single gold nanostars with (blue) and without (black) cytochrome c modification, where the dash line is an ideal CV. The electrolyte is 70 mM PBS (pH 7.0) and the scan rate is 0.1 V s⁻¹. **d** Scattering imaging of gold nanostars in **b**, where open circles indicate the location of gold nanostars without cytochrome c modification. Scale bar is 5 μm

in "Methods" section), which can be expressed by

$$i = \frac{neA\pi^{1/2}}{K_a} BL^{-1}\left\{s^{1/2}\overline{\Delta I/I}(s)\right\}, \quad (3)$$

where $B$ is $\left(z_O D_O^{-1/2} - z_R D_R^{-1/2}\right)^{-1}$, where $z_O$ and $z_R$ are the charges of the oxidized and reduced molecules, $D_O$ and $D_R$ are the diffusion coefficients of the redox species. $n$ is the number of electrons involved in one redox reaction, $e$ is the elementary charge, $K_a$ defines the adsorption of redox molecules, $L^{-1}$ is the inverse Laplace transform, and $\overline{\Delta I/I}(s)$ is the Laplace transform of $\Delta I/I$.

GEM allows for imaging the local faraday current density by performing Eq. (3) to scattering image sequence. To demonstrate

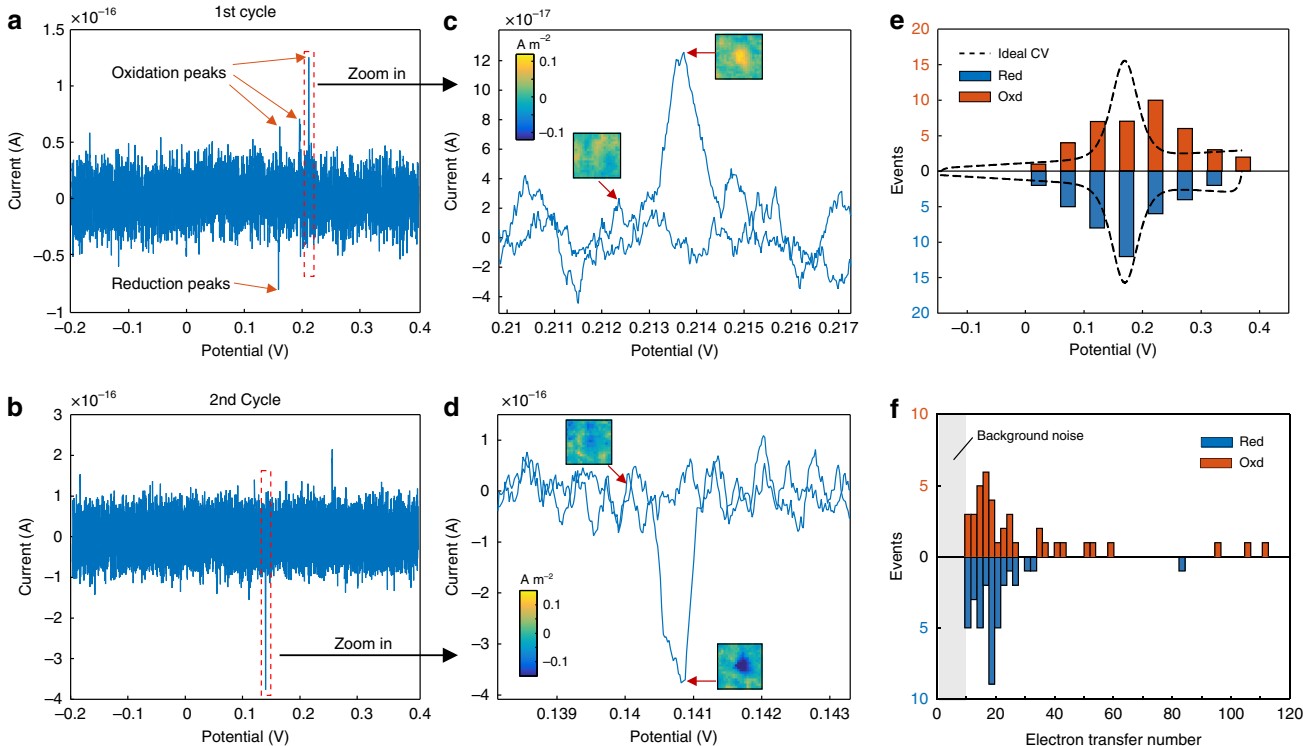

**Fig. 6** Stochastic electron transfer events measurement in cytochrome c. The first **(a)** and second **(b)** cycle of cyclic voltammograms of a single gold nanostar with cytochrome c modification. The electrolyte is 70 mM PBS (pH 7.0) and the scan rate is 0.01 V s$^{-1}$. **c, d** Magnifications of the region marked with dash squares in **a** and **b**, respectively. Insets showing current density images (1 by 1 μm) of the gold nanostar at corresponding potentials. **e** Histograms showing the distribution of single oxidation (red) and reduction (blue) events with the applied potential. The ideal CV of cytochrome c (dash line) is also shown. **f** Histograms showing the distribution of the electron transfer number of oxidation (red) and reduction (blue) events

it, we studied the cyclic voltammograms (CV) of 1 mM Fe(CN)$_6^{3-/4-}$ in 0.1 M KNO$_3$ with conventional electrochemical method and GEM, simultaneously. Figure 3a–d show snapshots of current density movie (Supplementary Movie 1) at different potentials. At −0.6 V, far away from the redox potential, the current density is near zero everywhere (Fig. 3a). When potential increases, oxidation of Fe(CN)$_6^{4-}$ takes place and a growing current density is observed where GNS are located, which reaches maximum at 0.12 V (Fig. 3b). As the potential cycles back, the current is inverted, attributed to the reduction of Fe(CN)$_6^{3-}$, and the maximum negative current is located at 0.02 V (Fig. 3c). The current eventually disappears when the potential cycles back to −0.6 V. The big contrast makes it possible to exclusively study electron transfer reactions at a single GNS without interference of graphene around, similar to ultramicroelectrodes.

Figure 3e displays CVs of multiple GNS (labeled with 1–5) extracted from the current density movie, and similar shapes are presented among these GNS in spite of some small deviations in the amplitudes of oxidation and reduction peaks due to their heterogeneity. We calculate the averaged CV of these GNS and compare it with the graphene area and the conventional CV recorded with a potentiostat (Fig. 3f). The CV of graphene area is indeed in good agreement with the conventional CV. Surprisingly, the CV of GNS shows ~10 times larger peak current density and a higher signal-to-noise ratio than that of graphene. We attribute it to the higher surface area of GNS, which offers more adsorption sites for reactive molecules. Moreover, the peak separation ($\Delta E_p$) of GNS (100 mV) is much closer to the ideal CV than that of the conventional CV (250 mV). It is well known that noble metals will give better electron transfer kinetics than graphene surfaces for inner-sphere redox couples, driven by the local density of states of the electrode near its Fermi level and the

reorganization energy of the molecules[43,44]. Such better electron transfer kinetics induces a faster accumulation of charges on the surface of GNS. As a result, a faster change in the scattering intensity is observed.

We now examine the anti-interference performance and the detection limit of GEM. The main interference chiefly arises from the change in refractive index, which is commonly the detection signal of other optical electrochemical strategies, such as P-ECi[15]. As shown in Fig. 4a, when we add 1% (v/v) ethanol to the electrolyte, the scattering intensity of a single GNS keeps steady, though a small fluctuation is induced by injection. For comparison, the total internal reflection intensity of the same area is recorded, which shows a stepwise decrement following the change in refractive index (Fig. 4b). Thus, the contribution of refractive index to GEM can be neglected. The current detection limit of GEM is determined by the background charging current noise level. As shown in Supplementary Fig. 2, the noise level is evaluated to be $7.2 \times 10^{-4}$ A m$^{-2}$. Thus, we estimate a current detection limit of $4.5 \times 10^{-18}$ A at a single GNS, since the near-field scattering cross section of GNS is $2.10 \times 10^{-15}$ m$^2$ (see "Methods" section). Such attoampere scale detection limit offers a considerable advance for electrochemical detection methods, which is currently in the picoampere to nanoampere range[13,15].

**Measuring electron transfer of cytochrome c molecules**. Measuring electron transfer at single-molecule level demands for both high spatial resolution and sensitivity, and the attoampere scale detection limit allows GEM to meet these demands. We take a redox protein cytochrome c for example to demonstrate this capability. Cytochrome c is adsorbed on 3-mercaptopropionic acid-modified GNS via the electrostatic attraction[18]. Scanning transmission electron microscopy (STEM) imaging of adsorbed

proteins is shown in Fig. 5a. The dash line describes the outline of a single GNS with sharp tips, which is coated by a uniform 2-nm-thick uranyl acetate negative staining layer (see "Methods" section). Multiple bulges are located at the two tips in the focal plane, representing adsorbed cytochrome c molecules. Small bulges (~3 nm) reveal individual molecules and big bulges (~6 nm) are induced by aggregates of several molecules (more examples in Supplementary Fig. 3). We count the number of cytochrome c molecules at each tip of at least 20 GNS, and found a concentrated distribution of histograms in range from 0 to 3 molecules (Supplementary Fig. 4). In consideration that only half area of a tip is visible, the amount of cytochrome c molecules at each tip is estimated to be 0–6. Thus a single GNS with around 8 tips should have 0–48 cytochrome c molecules. Measuring the electron transfer of such few cytochrome c molecules is barely to be achieved by current electrochemical technologies. For GEM, when electron transfer reactions take place on the graphene electrode without any diffusion, the current density could be simply measured by (details in "Methods" section)

$$i = Ae \frac{d(\Delta I/I)}{dt}. \tag{4}$$

Figure 5b shows the current image of multiple GNS at 0.16 V where the current reaches the maximum during a continuous cycling. Some of them have been immobilized with cytochrome c, demonstrating the clear contrast. However, the rest GNS with only 3-MPA modification, whose positions are marked by open circles (Fig. 5d), show no contrast. The CV of cytochrome c adsorbed on a single GNS (labeled with 1) is demonstrated in Fig. 5c (blue line), where a pair of well-defined reduction and oxidation peaks are found at around 0.16 V. The shape and peak current are similar to the theoretic CV (dash line) of a fully reversible one electron transfer reaction for 60 redox molecules immobilized on the graphene electrode[38]. In remarkable contrast, the CV of GNS without cytochrome c (labeled with 6) only shows background noise (Fig. 5c, black line) because the charging current is decreased by 3-MPA modification[18].

Further insight to the fast electron transfer events can be obtained by increasing the frame rate to 500 Hz, while the potential scan rate is set to be 10 mV s$^{-1}$ in order to reduce the charging background. We measure and compare the first (Fig. 6a) and second (Fig. 6b) CV cycles of a GNS with cytochrome c. Intriguingly, the broad reduction and oxidation peaks of cytochrome c become discrete spikes (magnifications in Fig. 6c, d), which are assigned to individual reduction and oxidation events. Deviations in the amplitudes reveal the different amount of electrons transferred in each event. Furthermore, spikes in different cycles occur at different potentials near the standard redox potential of cytochrome c during successive cycles, even at the same GNS. We attribute such stochastic spikes to dynamic states of cytochrome c molecules, arising from the lateral molecular interaction, variation in redox-site/electrode electronic coupling, or microenviromental variance[45]. To investigate whether the stochastic spikes from single electron transfer events can reproduce the ideal CV, we measure CVs of abundant GNS (Supplementary Fig. 5). As shown in Fig. 6e, histograms of reduction (blue) and oxidation (red) events both show distributions near the standard redox potential. The good correlation reveal that the apparent CV is the statistical result of stochastic electron transfer events.

We calculate the electron transfer number of these reduction and oxidation events to estimate the number of cytochrome c at a GNS. As shown in Fig. 6f, histograms of the electron transfer number during reduction (blue) and oxidation (red) events both show concentrated distributions in range from 10 to 30 electrons,

despite a rare distribution up to 115 is also observed. Note that ten electrons is the detection limit of our method due to the background noise level. Thus, the possibility of reduction and oxidation events with <10 transferred electrons should not be excluded. That is, one reduction or oxidation event involves varying number of cytochrome c, predominantly ranging from several to dozens of molecules, matching the number measured with STEM images.

## Discussion

In summary, we have proposed a universal electrochemical microscopy GEM based on the Fermi level-responsive optical conductivity of graphene to attain an ultrasensitive electron transfer measurement. Instead of measuring the current in circuit or the change in refractive index, GEM directly determines the change in local charge density. Ultrasensitive detection of local faraday current makes it possible to trace the electron transfer process in cytochrome c at single-molecule level. Although the detection limit of GEM is excellent relative to other electrochemical detection methods, the speed could be further improved by using high speed camera to trace individual dynamic electron transfer processes, which is in the range of nanoseconds to microseconds. Besides, the gold nanostars used here could be replaced by other scattering nanoparticles, such as Ag, TiO$_2$, and Au-Pt alloy nanoparticles. Since electron transfer reactions always involve variation in charge density, GEM provides a universal tool for study in many fields, such as basic chemical and biological phenomena, molecular electronics and energy materials. For example, *Shewanella* species use a direct electron transfer mechanisms to produce electricity through outer-membrane cytochrome c[46], and it is of great importance to exclusively study such process apart from electron shuttle-based indirect electron transfer mechanisms.

## Methods

**Measuring the optical conductivity of graphene.** In this work, a white light is used for imaging the scattering of graphene. Optical conductivity $\sigma(\omega)$ of graphene at frequency in visible region, where the interband transitions dominate, can be calculated simply by including the Fermi–Dirac distribution[30,34]:

$$\sigma(\omega) = \frac{1}{2}\sigma_0\left[\tanh\left(\frac{\hbar\omega + 2E_F}{4k_BT}\right) + \tanh\left(\frac{\hbar\omega - 2E_F}{4k_BT}\right)\right], \tag{5}$$

where $E_F$ is graphene's Fermi level, which is tuned by carrier density $n_c$ by $E_F = \hbar v_F(\pi n_c)^{1/2}$, $\sigma_0$ is the quantum conductivity defined as $e^2/4\hbar$. When potentials in range from $-0.8$ to $1.0$ V are applied, we measure $\sigma(\omega)$ of the graphene layer with corresponding transmittance $T$ by:

$$T = \sqrt{\frac{\epsilon_2}{\epsilon_1}}\frac{4(\epsilon_1\epsilon_0)^2}{\left[(\sqrt{\epsilon_1\epsilon_2} + \epsilon_1)\epsilon_0 + N\sqrt{\epsilon_1}\sigma(\omega)/c\right]^2}, \tag{6}$$

where $N$ is the number of layers, $\epsilon_0$, $\epsilon_1$, and $\epsilon_2$ are the vacuum permittivity, relative permittivities of medias below and above the graphene layer, respectively. In our work, the graphene layer is placed between glass ($\epsilon_1 = 2.25$) and the electrolyte ($\epsilon_2 = 1.77$).

The transmittance of graphene is measured with the microscope setup shown in Fig. 1b, except the normal incident light source. We move the area of interest to the border of the graphene layer, and measure the reflected intensity of the graphene ($I_{\text{graphene}}$) and the glass area ($I_{\text{glass}}$). The transmittance is calculated by:

$$T = \frac{I_{\text{graphene}}}{I_{\text{glass}}}. \tag{7}$$

Combining Eqs. (6) and (7), we have the local optical conductivity of the graphene at different potentials (Fig. 2a, black line).

**Measuring electrochemical current from scattering intensity.** We have shown that the charging current density can be measured by eq. (2). Now we focus on the contribution of $n_{ET}$ to describe electron transfer reactions. Electron transfer-induced charge density $q^{\text{ET}}$ can be expressed in terms of the oxidized and reduced product concentrations, $C_O$ and $C_R$, and given by

$$q^{\text{ET}}(t) = en_{ET} = FK_a\left[z_O C_O(0,t) + z_R C_R(0,t) - z_O C_O^0 - z_R C_R^0\right], \tag{8}$$

where $F$ is Faraday constant, $K_a = C_{\text{surf}}(t)/C(0,t)$ defines the adsorption of redox

molecules, $z_O$ and $z_R$ are the charges of oxidized and reduced molecules, $C_O^0$ and $C_R^0$ are the concentrations of oxidized and reduced molecules in bulk solution, respectively.

Conventional electrochemical methods measure current density versus potential or time, which is related to $C_O(0,t)$ and $C_R(0,t)$ [15,38]:

$$i(t) = -nFD_O \frac{\partial C_O(0,t)}{\partial x} = nFD_R \frac{\partial C_R(0,t)}{\partial x}, \qquad (9)$$

where $n$ is number of electrons transferred per reaction, and $D_O$ and $D_R$ are the diffusion coefficients of oxidized and reduced molecules, respectively.

During a redox reaction, the diffusion equation of oxidized species obeys Fick's laws:

$$\frac{\partial C_O(x,t)}{\partial t} = D_O \frac{\partial^2 C_O(x,t)}{\partial x^2}, \qquad (10)$$

where only the diffusion in vertical direction is considered because of the thin diffuse layer.

By performing Laplace transform on Eq. (10), we have

$$\bar{C}_O(x,s) = s^{-1} C_O^0 + A'(s) \exp[-(s/D_O)^{1/2} x], \qquad (11)$$

where $A'(s)$ is a function to be determined from boundary conditions at the electrode surface. To relate the concentrations to current density, we perform Laplace transform on Eq. (9) and then combine it with Eq. (11), leading to

$$\bar{C}_O(0,s) = s^{-1} C_O^0 + (nFD_O^{1/2})^{-1} s^{-1/2} \bar{i}(s) \qquad (12)$$

and a similar relation could be obtained for the reduced species

$$\bar{C}_R(0,s) = s^{-1} C_R^0 - (nFD_R^{1/2})^{-1} s^{-1/2} \bar{i}(s). \qquad (13)$$

Combining Eqs. (8), (12), and (13), we have

$$q^{ET}(t) = en_{ET} = \frac{K_a}{n\pi^{1/2}} \left( z_O D_O^{-1/2} - z_R D_R^{-1/2} \right) \int_0^t i(\tau)(t-\tau)^{-1/2} d\tau. \qquad (14)$$

Substituting Eq. (1) into Eq. (14), we have

$$\Delta I/I(t) = \frac{K_a}{Aen\pi^{1/2}} \left( z_O D_O^{-1/2} - z_R D_R^{-1/2} \right) \int_0^t i(\tau)(t-\tau)^{-1/2} d\tau. \qquad (15)$$

Performing Laplace transform on Eq. (15), and we have

$$\overline{\Delta I/I}(s) = \frac{K_a}{Aen\pi^{1/2}} \left( z_O D_O^{-1/2} - z_R D_R^{-1/2} \right) s^{-1/2} \bar{i}(s). \qquad (16)$$

Thus, the faraday current density can be given by:

$$i(t) = \frac{neA\pi^{1/2}}{K_a} \left( z_O D_O^{-1/2} - z_R D_R^{-1/2} \right)^{-1} L^{-1} \left\{ s^{1/2} \overline{\Delta I/I}(s) \right\}, \qquad (17)$$

where $L^{-1}$ is the inverse Laplace transform. For $Fe(CN)_6^{3-/4-}$, $n$ is 1, $z_O$, and $z_R$ are $-3$ and $-4$, and $D_O$ and $D_R$ are $7.2 \times 10^{-10}$ and $6.67 \times 10^{-10}$ m$^2$ s$^{-1}$, respectively[47]. $K_a$ is determined to be $1 \times 10^{-7}$ m (see below).

When electrochemical reactions occur only on the graphene electrode, such as redox reactions of adsorbed proteins, the current density can be simply given by[17]:

$$i(t) = nF \frac{dC_O(t)}{dt} = -nF \frac{dC_R(t)}{dt}, \qquad (18)$$

where $C_O(t)$ and $C_R(t)$ are surface concentrations of the oxidized and reduced products. Combining Eqs. (6) and (18), and we have

$$i(t) = e \frac{dn_{ET}}{dt} = Ae \frac{d(\Delta I/I)}{dt}. \qquad (19)$$

Note that Eq. (19) shows the same express as Eq. (2). That is to say, the faraday current density could be measured together with the charging current in this model.

**Calibration of $K_a$.** To calibrate the adsorption constant $K_a$, defined as $K_a = C_{surf}(t)/C(0,t)$, we measure the change in $\Delta I/I$ of gold nanostars induced by the addition of 1 mM $Fe(CN)_6^{3-/4-}$. To simplify the model, a $+0.6$ V potential is applied, which is more positive than the standard oxidation potential (Fig. 3e). The positive potential leaves almost only $Fe(CN)_6^{3-}$ in the diffusion layer (hundreds of micrometers thick) near the graphene surface. The electrolyte is 0.1 M $KNO_3$, thus the addition of 1 mM $K_3Fe(CN)_6$ will not affect the ion strength and the electrical double layer. Thus $\Delta I/I$ is only contributed by the adsorption of $Fe(CN)_6^{3-}$ ions, and $K_a$ is expressed by:

$$K_a = \frac{\Delta n_c}{z_O N_a C(0,t)} = \frac{A\Delta(\Delta I/I)}{z_O N_a C_O(0,t)}, \qquad (20)$$

where $N_a$ is Avogadro's constant, and $K_a$ is found to be $1 \times 10^{-7}$ m.

**Calculation of near-field scattering cross section.** The near-field scattering cross section $C_{sca}$ of a gold nanostar, which determines the scattering intensity and therefore the current at single gold nanostars, is calculated by the effective

polarizability by[32,33]:

$$C_{sca} = \frac{k^4 |\alpha_\perp^{eff}|^2}{6\pi} \qquad (21)$$

where $k = 2\pi/\lambda$ is the wave number, and $\alpha_\perp^{eff}$ is the effective polarizability, governed by:

$$\alpha_\perp^{eff} = \frac{\alpha(1+\beta)}{1 - \frac{\alpha\beta}{16\pi(r+d)^3}}, \qquad (22)$$

$$\beta = \frac{\varepsilon_g - 1}{\varepsilon_g + 1}, \alpha = 4\pi r^3 \frac{\varepsilon_{Au} - 1}{\varepsilon_{Au} + 1},$$

where $\varepsilon_g$ and $\varepsilon_{Au}$ are the dielectric constant of graphene and gold, respectively and $r = 40$ nm is the radius of gold nanostars, which is considered as gold nanospheres for simplification because of the similar scattering cross-section of two types of nanoparticles (Supplementary Fig. 6). According to Eqs. (21) and (22), $C_{sca}$ is calculated to be $2.10 \times 10^{-15}$ m$^2$.

**Chemicals and general techniques.** Poly(methyl methacrylate) and gold etchant were purchased from Sigma-Aldrich (Shanghai, China). Gold nanostars (80-nm core diameter) and gold nanospheres (80-nm diameter) were purchased from NanoSeedz Ltd CVD Graphene (3–5 layers) on copper foil is purchased from Nanjing XFNANO Materials Tech Co., Ltd Absolute ethanol, acetone, cysteamine, 3-mercaptopropionic acid, and purified bovine heart cytochrome c were purchased from Aladdin Reagent Inc. PBS was purchased from Nanjing KeyGen Biotech. Co. Ltd. All other reagents are of analytical grade. Ultrapure water with a resistivity of 18.2 MΩ cm was produced using a Milli-Q apparatus (Millipore) and used in the preparation of all solutions. Cover slides were purchased from Thorlabs Co., Ltd PDMS was prepared using Sylgard 184, Dow Corning. Copper etchant was prepared by dissolving 10 g $CuSO_4$ in 50 mL deionized water and 50 mL 37% hydrochloric acid.

UV-vis spectra were recorded on a UV-1750 spectrophotometer (Shimadzu, Kyoto, Japan). Scanning transmission electron micrographs were captured on a JEOL 2800 transmission electron microscope. Dark-field images and spectra measurements were carried out on Nikon Ti-E microscope. A broadband light source (EQ-99XFC LDLS, Energetiq Technology) was used for incident illumination. True-color dark-field images are captured by a color-cooled digital camera (DS-RI1, Nikon), and the scattering spectra of single nanoparticles was measured by a monochromator (Acton SP2300i, PI) equipped with a spectrograph CCD (PIXIS 400BR_excelon, PI) and a grating (grating density: 300 L mm$^{-1}$; blazed wavelength: 500 nm). The conventional electrochemical experiments were carried out on a potentiostat (ACFBP1, Pine Research Instrumentation).

**Preparation of gold nanostars.** Gold nanostars were prepared via a typical seed-mediated growth process. The seed solution is prepared by dissolving 0.25 mL citrate (0.01 M) and 0.125 mL $HAuCl_4$ (0.01 M) to 9.625 mL DI water, followed by the addition of 150 μL fresh cold $NaBH_4$ (0.01 M). The solution is then shaken for 3 h. To prepare the growth solution, 42.75 mL Tetradecyltrimethylammonium bromide (0.1 M), 1.8 mL $HAuCl_4$ (0.01 M), 270 μL $AgNO_3$ (0.01 M), and 300 μL ascorbic acid (0.1 M) were mixed. Finally, 60 μL seed solution was added to the growth solution, followed by incubation at room temperature overnight.

**Fabrication of the electrochemical cell.** A 47-nm-thick gold film was coated on cover slide, followed by treatment of gold etchant for 1 min in the center. The remaining gold film was used for connection between the graphene and the potentiostat. A CVD graphene sample was transferred onto the etched hole of the gold substrate with a PMMA-mediated approach. Simply, a layer of PMMA was spin-coated onto the graphene, and the metal below it was etched away completely. The PMMA/graphene stack was then transferred onto the Au surface. After the graphene was transferred onto the gold substrate, the PMMA layer was dissolved and removed by acetone. An electrochemical cell (with 3.5 mm inner diameter) made of PDMS on was placed on top of the graphene sample, and $KNO_3$ or PBS solution was used as electrolyte. The potential of graphene was controlled with respect to Ag/AgCl reference electrode with the potentiostat using a platinum wire as counter electrode. Gold nanostars were then deposited on the graphene for following experiments. To immobilize individual gold nanostars on the graphene, 100 μL of ultrapure water was added into the cell, followed by the addition of 10 μL 20 pM gold nanostars. After sedimentation for 10 min, excess gold nanostars were removed by pipet and the electrochemical cell was thoroughly rinsed with the ultrapure water.

**Optical measurement and imaging processing.** For dark-field scattering imaging, the electrochemical cell was placed on the 100× oil immersion objective (NA = 1.49) equipped by a Nikon Ti-E inverted microscope. A barrier was placed at the back focus plane of the objective to stop the reflected light and only the scattering light was directed to a CMOS camera (AVT Pike F-032B). For total internal reflection imaging, the barrier was removed. In order to calculate the relative scattering change (ΔI/I), the background scattering was initially removed

by subtracting the scattering intensity far away from the Dirac point. The pure potential dependent scattering was then normalized by dividing the scattering intensity at the Dirac point. The relative scattering change at each pixel was processed to produce a current density image of the surface. The frame rate is 10 Hz for all optical measurements except that shown in Fig. 6, Supplementary Fig. 5 (500 Hz).

**Charge doping and cytochrome c modification**. The doping charges of graphene were modulated by adding 0.1 M PBS (pH from 2.5 to 10) to the electrolyte (0.1 M KNO$_3$). For cytochrome c modification, gold nanostars were immersed in 5 mM 3-mercaptopropionic acid for 2 h and subsequently 50 µM cytochrome c for 1 h. In order to observe the negative contrast of single cytochrome c molecules with STEM, gold nanostars with cytochrome c modification were incubated in 2% uranyl acetate for 2 s.

## Data availability
The data and computer codes supporting the findings of this study are available from the authors upon reasonable request.

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

## Acknowledgements
This research is supported by the National Natural Science Foundation of China (Grants nos. 21834004, 21327902, 21427807, and 21605081), the Natural Science Foundation of Jiangsu Province (Grants no. BK20160638), the International Cooperation Foundation from the Ministry of Science and Technology (Grants no. 2016YFE0130100), the Fundamental Research Funds for the Central Universities (Grants no. 020514380173), and the Excellent Research Program of Nanjing University (Grants no. ZYJH004).

## Author contributions
Z.C. and J.J.Z. conceived the study. Q.X., P.X. and M.W. performed the experiments. Z.C. and X.C. built the microscope setup. Z.C., Q.X. and J.J.Z. designed the experiments. J.R.Z. and H.Y.C. advised the experiments. Z.C. analyzed the data and wrote the paper.

## Additional information

**Competing interests:** The authors declare no competing interests.

