## [Peer Review File · Nature Communications]

Reviewers' comments:

Reviewer #1 (Remarks to the Author):

The manuscript reports very interesting graphene-based electrochemical microscopy (GEM) technique that makes an advance in the detection limit for electron transfer. The paper is well written and is recommended publishing after a very minor revision.

1. Fig 3, the current unit A/m² looks too large. Please explain.
2. Is it possible to use COMSOL for Page 12 instead of Laplace transform?
3. In the movie, the current unit A/m² looks too large.

Reviewer #2 (Remarks to the Author):

Comments –

ABSTRACT: The authors are correct in saying that electrochemical measurements do not normally go below the nanoamp/picoamp current range – but there are some advanced works that do make efforts to resolve currents at lower ranges.

Fig 1(a) – it would help to give some idea of dimensions of the parts of the electrochemical cell . Fig 1(c) – the “nanostar” does not seem to have a very regular shape, from the TEM presented. How regular are these structures? SI Fig3 also confirms this concern. Also, the irregular shape causes a problem as the optical model seems to assume that the Au objects are spherical – so how good is this approximation? I did not get how the nanostars are made (I found in the Methods section that they are a commercial product, but some information on their preparation would help). Also, there was no detail of how they are immobilised on the graphene electrode – how is control over their separation/loading achieved, if at all? Finally, the graphene is not monolayer sample but 3-5 layers thick (according to Methods) – does the exact thickness matter?

Fig 2 – I assume the potentials quoted in the text when discussing this Figure relate to the reference electrode used?

Fig 2 (d) would be more convincing if the authors could show that the shift in potential, with different adsorbates, is sensitive to the concentration (adsorption) of the molecules added.

In the derivation of the current response, it is not clear to me why the adsorption equilibrium is required to calculate the Faradic charge transferred – why do the molecules have to be adsorbed on the electrode to contribute to the current?

Some references for the diffusion coefficients quoted for the ferri/ferro cyanide species (Methods) should be given.

More explanation for the statement on line 318, about the effect of the positive potential, is needed. It depends on the electrolyte strength, the distance considered relative to the electrode and the position of the applied potential vs. the standard reduction potential of this molecule.

How do the authors deconvolute the effects of surface chemistry (Fig 3)? It is well known that noble metals will give better electron transfer kinetics than graphene surfaces for inner-sphere redox couples such as ferri/ferro-cyanide, so how are these effects (which are driven by the local density of states of the electrode at/near its Fermi level and the reorganisation energy of the molecules) accounted for in the optical approach? This would explain the change in peak separation (potential scale) of the CV, but not the change in current density seen – which can only be rationalized if it is assumed that a very different adsorption of the molecule occurs on the gold, is this the case?

The authors give a vague explanation on this point (line 175).

The explanation of the cytochrome c surface preparation is also poor – for example, there is no mention (Methods) of the uranyl staining layer, so it is not clear how it is introduced. The estimation of the number of cyt c molecules seemed highly approximate.

Fig 5(e) – which trace is which? The labelling is inadequate.

So, to summarize - some revisions are needed but

the work is interesting , however, and novel as far as I know. It raises intriguing questions – eg. what

is the time resolution of the technique? Is it possible to resolve stochastic processes, e.g. electrons transferring to/from a few protein molecules if the potential is rapidly oscillated either side of the standard reduction potential of the molecule?

Point-by-point response to the comments

Reviewer #1

General comment: The manuscript reports very interesting graphene-based electrochemical microscopy (GEM) technique that makes an advance in the detection limit for electron transfer. The paper is well written and is recommended publishing after a very minor revision.

Response: We thank the referee for reviewing our manuscript, affirming its clarity and comprehensibility, and for her/his suggestions on how to improve the manuscript. The provided comments and questions are responded point by point below.

Comment 1: Fig 3, the current unit A/m² looks too large. Please explain.

Response: The current unit A/m² equals to 1 $\mu\text{A}/\text{mm}^2$, which is a normal unit for current density. As shown in Fig. 3f, we compare the CV of GNS (red), graphene area (black) and the conventional CV (blue) recorded with a potentiostat in 1 mM $\text{Fe}(\text{CN})_6^{3-/4-}$. It is clear that the current density of graphene area matches the conventional CV quite well, while the CV of GNS shows ~ 10 times larger peak current density, attributed to the higher surface area of GNS.

Comment 2: Is it possible to use COMSOL for Page 12 instead of Laplace transform?

Response: We thank the referee's advice for the formula derivation. We use Laplace transform here to build the relationship between the faraday current density and the surface concentration of redox molecules, according to the classic method described in the book (*Bard, A. J. & Faulkner, L. R., Electrochemical methods: fundamentals and applications, Wiley, 1980*) and the reference (*Science, 327, 1363-1366*). To our knowledge, COMSOL is a useful software in the field of Numerical Simulation. It can help us generate a model for explanation of the

relationship between the current density and the optical intensity. Hence it should be an alternative way to attain the expression of faraday current. In this work, however, we need a classic way to simply get the relationship between the concentration and current, and then use it to convert the scattering intensity to the faraday current.

Comment 3: In the movie, the current unit A/m² looks too large.

Response: Snapshots of Supplementary Movie 1 are displayed in Fig. 3a-d, and the CVs in Fig. 3f were extracted from the movie. As described in the response to Comment 1, the current density of graphene area matched the conventional CV quite well, thus it is reliable.

Reviewer #2

Comment 1: ABSTRACT: The authors are correct in saying that electrochemical measurements do not normally go below the nanoamp/picoamp current range – but there are some advanced works that do make efforts to resolve currents at lower ranges.

Response: We thank the referee for reviewing our manuscript, affirming its clarity and comprehensibility, and for her/his suggestions on how to improve the manuscript. We agree with that there are some advanced works that do make efforts to resolve currents at lower ranges. Thus, we have replaced the word “fundamentally” by “normally” in the first sentence of abstract. In addition, we have added the following discussion to the introduction section (line 47).

“Many efforts have been made to resolve the current at lower ranges, such as supermicroelectrodes^{19,20} and surface-enhanced Raman spectroscopy^{21,22}, however, the detection limit is normally restricted in the picoampere to nanoampere range^{13,15}.”

Comment 2: Fig 1(a) - it would help to give some idea of dimensions of the parts of the electrochemical cell.

Response: According to the referee's advice, we have added dimensions of the parts of electrochemical cell to Fig. 1a, and the following description has been added to the caption (line 574).

“A graphene layer is transferred onto a gold-coated cover slide with a 4 mm-diameter hole in center, on which attaching a 3.5-mm-diameter PDMS electrochemical cell to avoid reactions on the gold film.”

Comment 3: Fig 1(c) – the “nanostar” does not seem to have a very regular shape, from the TEM presented. How regular are these structures? SI Fig3 also confirms this concern. Also, the irregular shape causes a problem as the optical model seems to assume that the Au objects are spherical – so how good is this approximation?

Response: We use gold nanostars as nanoelectrodes in this work because of its large surface area for redox molecules adsorption and broad scattering from visible to near-infrared region. They have uniform 80-nm-diameter cores and 6~10 tips outside that makes them look irregular. To investigate how good the approximation is, we immobilize 80-nm-diameter regular gold nanospheres and gold nanostars in the same area of graphene, successively. As shown in SI Fig. 6a, both of two types of gold nanoparticles show clear scattering spots. We measure the scattering intensity of 41 gold nanospheres and 41 nanostars, respectively. Histograms show the similar scattering intensity distribution of the two types of gold nanoparticles (SI Fig. 6b). That is, gold nanostars have a uniform scattering cross-section, which is close to that of gold nanospheres. Accordingly, we assume that the Au objects are spherical with 40 nm radius to simplify the calculation of the near-field scattering cross-section of gold nanostars. The correlated discussion has been added to the revised manuscript (line 349).

Comment 4: I did not get how the nanostars are made (I found in the Methods section that they are a

commercial product, but some information on their preparation would help). Also, there was no detail of how they are immobilised on the graphene electrode – how is control over their separation/loading achieved, if at all? Finally, the graphene is not monolayer sample but 3-5 layers thick (according to Methods) – does the exact thickness matter?

Response: We asked for the preparation procedure of gold nanostars, and the following content has been added to the Methods section (line 379).

“Gold nanostars were prepared via a typical seed-mediated growth process. The seed solution is prepared by dissolving 0.25 mL citrate (0.01 M) and 0.125 mL HAuCl₄ (0.01 M) to 9.625 mL DI water, followed by the addition of 150 μ L fresh cold NaBH₄ (0.01 M). The solution is then shaken for 3 h. To prepare the growth solution, 42.75 mL Tetradecyltrimethylammonium bromide (0.1 M), 1.8 mL HAuCl₄ (0.01 M), 270 μ L AgNO₃ (0.01 M) and 300 μ L ascorbic acid (0.1 M) were mixed. Finally, 60 μ L seed solution was added to the growth solution, followed by incubation at room temperature overnight.”

To immobilize individual gold nanostars on the graphene, 100 μ L of ultrapure water was added into the cell, followed by the addition of 10 μ L 20 pM gold nanostars. After sedimentation for 10 min, excess gold nanostars were removed by pipet and the electrochemical cell was thoroughly rinsed with the ultrapure water. Above procedures have been also added to Methods section (line 399).

The CVD graphene (3-5 layers) on copper foil is purchased from Nanjing XFNANO Materials Tech Co., Ltd. We asked the company for the exact thickness, the graphene might be 3-layer, 4-layer or 5-layer in different samples. However, the deviation of layer number will not affect the result. According to the Eq. (5), the optical conductivity of graphene is only determined by its Fermi level. We have discussed the linear relationship of relative scattering change ($\Delta I/I$) and the optical conductivity in the Results section (line 114), which consequently yields the correlation of graphene’s carrier density and $\Delta I/I$, according to Eq. (1).

Comment 5: Fig 2 - I assume the potentials quoted in the text when discussing this Figure relate to the reference electrode used? Fig 2 (d) would be more convincing if the authors could show that the shift in potential, with different adsorbates, is sensitive to the concentration (adsorption) of the molecules added.

Response: Fig 2 -Yes, all potentials in this manuscript are relative to the reference electrode. We have added the following statement to the revised manuscript (line 83).

“All potentials mentioned in this work are relative to reference.”

Fig 2 (d)-The shift in potential could be induced by charges doping in the graphene. We initially tried to use cysteamine and 3-mercaptopropionic acid for positive and negative doping. However, the Au-S bond-based adsorption is not easy to be controlled by the concentration of incubation molecules. To investigate the concentration-dependent shift in the potential, we use the hydroxyl and hydroxonium ions instead to modulate Fermi level of the graphene by doping positive and negative charges, respectively. Fig. 2d has been updated with the new data, and the following description has been added to the revised manuscript (line 125).

“In order to further understand the contribution of excess charges to the scattering intensity, we change the electrolyte pH from 2.5 to 10, leaving the graphene with excess negative and positive charged carriers, respectively⁴¹.”

Comment 6: In the derivation of the current response, it is not clear to me why the adsorption equilibrium is required to calculate the Faradic charge transferred – why do the molecules have to be adsorbed on the electrode to contribute to the current?

Response: The GEM method proposed in this work is based on the charge-dependent change in the optical conductivity of graphene. We have discussed the relationship of the charged carrier density in the graphene and the relative scattering change (Eq. (1)). To change the carrier density in this system, one has two common ways:

the electrical double layer gating and the charge doping. The electrical double layer gating effect has been illustrated in Fig. 2(a), which contributes to the charging current. To dope charges in the graphene, ions need to enter the compact layer, otherwise charges will be neutralized by the double layer. Thus, molecules have to be adsorbed on the electrode to contribute to the faraday current. We had tried other outer-sphere redox molecules that will not adsorb on the electrode surface, for example $\text{Ru}(\text{NH}_3)_6^{2+/3+}$, and no redox current can be observed by GEM.

Comment 7: Some references for the diffusion coefficients quoted for the ferri/ferro cyanide species (Methods) should be given.

Response: Correlated references have been given in the Methods (line 320).

Comment 8: More explanation for the statement on line 318, about the effect of the positive potential, is needed. It depends on the electrolyte strength, the distance considered relative to the electrode and the position of the applied potential vs. the standard reduction potential of this molecule.

Response: We have added the following explanation to the revised manuscript (line 332).

“To calibrate the adsorption constant K_a , defined as $K_a = C_{\text{surf}}(t)/C(0, t)$, we measure the change in $\Delta I/I$ of gold nanostars induced by the addition of 1 mM $\text{Fe}(\text{CN})_6^{3-/4-}$. To simplify the model, a +0.6 V potential is applied, which is more positive than the standard oxidation potential (Fig. 3e). The positive potential leaves almost only $\text{Fe}(\text{CN})_6^{3-}$ in the diffusion layer (hundreds of micrometers thick) near the graphene surface. The electrolyte is 0.1 M KNO_3 , thus the addition of 1 mM $\text{K}_3\text{Fe}(\text{CN})_6$ will not affect the ion strength and the electrical double layer. Thus $\Delta I/I$ is only contributed by the adsorption of $\text{Fe}(\text{CN})_6^{3-}$ ions, and K_a is expressed by:

$$K_a = \frac{\Delta n_c}{z_0 N_a C_0(0, t)} = \frac{A \Delta(\Delta I/I)}{z_0 N_a C_0(0, t)} \quad (20)$$

where N_a is Avogadro's constant, and K_a is found to be 1×10^{-7} m."

Comment 9: How do the authors deconvolute the effects of surface chemistry (Fig 3)? It is well known that noble metals will give better electron transfer kinetics than graphene surfaces for inner-sphere redox couples such as ferri/ferro-cyanide, so how are these effects (which are driven by the local density of states of the electrode at/near its Fermi level and the reorganisation energy of the molecules) accounted for in the optical approach? This would explain the change in peak separation (potential scale) of the CV, but not the change in current density seen – which can only be rationalized if it is assumed that a very different adsorption of the molecule occurs on the gold, is this the case?

The authors give a vague explanation on this point (line 175).

Response: We have added the following detail discussion about how to explain the change in peak separation and current density to the revised manuscript (line 171-179).

“Surprisingly, the CV of GNS shows ~10 times larger peak current density and a higher signal-to-noise ratio than that of graphene. We attribute it to the higher surface area of GNS, which offers more adsorption sites for reactive molecules. Moreover, the peak separation (ΔE_p) of GNS (100 mV) is much closer to the ideal CV than that of the conventional CV (250 mV). It is well known that noble metals will give better electron transfer kinetics than graphene surfaces for inner-sphere redox couples, driven by the local density of states of the electrode near its Fermi level and the reorganization energy of the molecules. Such better electron transfer kinetics induces a faster accumulation of charges on the surface of GNS. As a result, a faster change in the scattering intensity are observed.”

Comment 10: The explanation of the cytochrome c surface preparation is also poor – for example, there is no mention (Methods) of the uranyl staining layer, so it is not clear how it is introduced. The estimation of the number of cyt c molecules seemed highly approximate.

Response: The uranyl acetate was introduced as negative staining layer for STEM imaging to improve the poor contrast of cytochrome c molecules. We have added the following detail process to the Methods section in the revised manuscript (line 420).

“In order to observe the negative contrast of single cytochrome c molecules with STEM, gold nanostars with cytochrome c modification were incubated in 2 % uranyl acetate for 2 s.”

As the referee commented, the estimation of the number of cyt c molecules is highly approximate. We tried to image the GNS with STEM, but it’s hard to observe all attached cyt c molecules due to the shooting angle. Thus, what we do is to count the number of cyt c molecules on each tip, and then estimate the total number of molecules on a gold nanostar. To estimate the number of cyt c molecules more accurately, we count the number of cyt c molecules at each tip of at least 20 gold nanostars. The following results have been added to the revised manuscript (line 204).

“We count the number of cytochrome c molecules at each tip of at least 20 GNS, and found a concentrated distribution of histograms in range from 0 to 3 molecules (Supplementary Fig. 4). In consideration that only half area of a tip is visible, the amount of cytochrome c molecules at each tip is estimated to be 0 to 6. Thus a single GNS with around 8 tips should have 0 to 48 cytochrome c molecules.”

Comment 11: Fig 5(e) – which trace is which? The labelling is inadequate.

Response: Fig 5(e) and the correlated discussion were removed in the revised manuscript, while the new results have been added to resolve stochastic processes of electron transfer, according to the referee’s advice in

Comment 12. We tried to use the integration of CVs to describe the electron transfer accumulation. However, the high scan rate (0.1 V/s) induces a huge charging current background, and the low frame rate (10 Hz) cannot afford the capture of fast electron transfer processes, ranging from nanoseconds to microseconds. To solve it, we increase the frame rate to 500 Hz, while the potential scan rate is set to be 10 mV/s. Detail results are shown in the response to Comment 12.

Comment 12: So, to summarize - some revisions are needed but the work is interesting, however, and novel as far as I know. It raises intriguing questions – eg. what is the time resolution of the technique? Is it possible to resolve stochastic processes, e.,g electrons transferring to/from a few protein molecules if the potential is rapidly oscillated either side of the standard reduction potential of the molecule?

Response: We thank the referee's advice about the potential application in resolving stochastic processes.

Actually, we have already got some relevant results in recent experiments. We increased the frame rate to 500 Hz, which could be further improved by using high speed camera. Intriguingly, we found the discrete spikes instead of broad redox peaks in the CV, which are assigned to individual fast electron transfer events. We thought it is more suitable for estimation of the amount of electron transfer than the integration of low-frame-rate CVs, so we replaced the last paragraph in Results section by the following discussion in the revised manuscript (line 224).

“Further insight to the fast electron transfer events can be obtained by increasing the frame rate to 500 Hz, while the potential scan rate is set to be 10 mV/s in order to reduce the charging background. We measure and compare the first (Fig. 6a) and second (Fig. 6b) CV cycles of a GNS with cytochrome c. Intriguingly, the broad reduction and oxidation peaks of cytochrome c become discrete spikes (magnified in Fig. 6c-d), which are assigned to individual reduction and oxidation events. Deviations in the amplitudes reveal the different amount

of electrons transferred in each event. Furthermore, spikes in different cycles occur at different potentials near the standard redox potential of cytochrome c during successive cycles, even at the same GNS. We attribute such stochastic spikes to dynamic states of cytochrome c molecules, arising from the lateral molecular interaction, variation in redox-site/electrode electronic coupling, or microenvironmental variance⁴⁴. To investigate whether the stochastic spikes from single electron transfer events can reproduce the ideal CV, we measure CVs of abundant GNS (Supplementary Fig. 5). As shown in Fig. 6e, histograms of reduction (blue) and oxidation (red) events both show distributions near the standard redox potential. The good correlation reveals that the apparent CV is the statistical result of stochastic electron transfer events.

We calculate the electron transfer number of these reduction and oxidation events to estimate the number of cytochrome c at a GNS. As shown in Fig. 6f, histograms of the electron transfer number during reduction (blue) and oxidation (red) events both show concentrated distributions in range from 10 to 30 electrons, despite a rare distribution up to 115 is also observed. Note that 10 electrons is the detection limit of our method due to the background noise level. Thus, the possibility of reduction and oxidation events with less than 10 transferred electrons should not be excluded. That is, one reduction or oxidation event involves varying number of cytochrome c, predominantly ranging from several to dozens of molecules, matching the number measured with STEM images.”

REVIEWERS' COMMENTS:

Reviewer #1 (Remarks to the Author):

I was reviewer 1 and recommended the manuscript being published after a very minor revision.

Anyway, I am satisfied with the authors' reply. The paper is ready to go in press.

Reviewer #2 (Remarks to the Author):

Comments – Is “supermicroelectrode” (added to revised Introduction) a widely accepted term? It is new to me

Some mistakes in English – e.g. p4, line 64 “coverts” should say “converts”

Precision of the quoted detection limit, strange that it's quoted to three significant figures?

I do not understand the modification made to Fig 2. Fig 2(c) is a cartoon that shows n-doping of the graphene at low pH, although an excess of anions is shown to be adsorbed, I do not get why this should be. Are the anions hydroxide? If so, this does not make sense as we are dealing with low pH. The converse arguments hold for the p-doping picture. The physical basis of this picture does not make sense to me. The authors cite ref 41 as justification for their work, but this has proven to be controversial with some works (e.g. Nano Lett.2011,11, 3597-3600) offering completely different observations.

I think the statement about the faster electron transfer kinetics on metals, while true and valid, should be backed up with some references.

I also still do not understand the basis of the method, perhaps this is my misunderstanding, but the process is not clear to me. Equation (8) implies that an adsorption of the redox molecule is required for the method to work – is this correct and what sort of adsorption process is it? Is it a transient adsorption taking place during the electron transfer event only?

Reviewer #3 (Remarks to the Author):

The authors have addressed all the questions and i would like to recommend the work for publication.

Point-by-point response to the comments

Reviewer #2

Comment 1: Is “supermicroelectrode” (added to revised Introduction) a widely accepted term? It is new to me.

Response: We thank the referee for reviewing our manuscript, affirming its clarity and comprehensibility, and for her/his suggestions on how to improve the manuscript. We agree with that the expression of related terms is not standardized. Thus, we have replaced the word “supermicroelectrode” by “ultramicroelectrode”, a widely accepted term, in the introduction (line 49).

Comment 2: Some mistakes in English – e.g. p4, line 64 “coverts” should say “converts”.

Response: We thank the referee’s advice for the spelling mistake. We have replaced the word “coverts” by “converts” (line 71).

Comment 3: Precision of the quoted detection limit, strange that it’s quoted to three significant figures?

Response: We thank the referee for the advice on the digital specification. The significant figures of the detection limit should be determined by the scattering intensity (I) and the sampling interval (t), because other data numbers were all calculated from theoretic simulations (Equation 2). The scattering intensity is recorded with a CMOS camera, changing from 10000 to 16000, shows five significant figures. The sampling interval, while the frame rate of the camera is set to be 10 Hz, is 0.10 s. That is, the detection limit should have two significant figures. Thus, we have changed the detection limit to 4.5×10^{-18} A (line 186).

Comment 4: I do not understand the modification made to Fig 2. Fig 2(c) is a cartoon that shows n-doping of the graphene at low pH, although an excess of anions is shown to be adsorbed, I do not get why this should be. Are the anions hydroxide? If so, this does not make sense as we are dealing with low pH. The converse arguments hold for the p-doping picture. The physical basis of this picture does not make sense to me. The authors cite ref 41 as justification for their work, but this has proven to be controversial with some works (e.g. Nano Lett. 2011, 11, 3597-3600) offering completely different observations.

Response: We are glad to explain the cartoon in Fig. 2 (c) in detail. The n-doping of the graphene at

low pH is not induced by anions adsorption but the accumulation of H_3O^+ ions. As pH increases, p-doping increases with the adsorption of OH^- ions. As a result, the charge-neutral point of gate voltage will shift towards more positive potential with pH increasing. Such result has been reported in many works, including Ref 41 and Nano Lett. 2011, 11, 3597-3600.

The reference (Nano Lett. 2011, 11, 3597-3600) introduced by the referee claimed that the large range of pH-induced gate shifts observed in the previous literature reflects the quality of graphene. Defective graphene, where free bonds exist on the surface, shows a large shift, whereas high-quality graphene with no dangling bonds, show no shifts to pH. Ref 41 (J. Am. Chem. Soc. 2008, 130, 14392–14393) also supported this theory, and attributed the pH-sensitivity of graphene to surface-bond OH species. The CVD graphene film on copper foil used in this work is produced by ACS Material, which might expose abundant free bonds on the surface.

Comment 5: I think the statement about the faster electron transfer kinetics on metals, while true and valid, should be backed up with some references.

Response: Correlated references have been given to back up the faster electron transfer kinetics on metals (line 174). [J. Am. Chem. Soc. 2016, 138, 13975-13984, and J. Am. Chem. Soc. 2011, 133, 4,762-764.]

Comment 6: I also still do not understand the basis of the method, perhaps this is my misunderstanding, but the process is not clear to me. Equation (8) implies that an adsorption of the redox molecule is required for the method to work – is this correct and what sort of adsorption process is it? Is it a transient adsorption taking place during the electron transfer event only?

Response: Yes, an adsorption of the redox molecule is required. In conventional electrochemistry, the solution side of the double layer is thought to be made up of several "layers." That closest to the graphene, the compact layer, contains solvent molecules and sometimes other species (ions or molecules) that are said to be specifically adsorbed, such as $\text{Fe}(\text{CN})_6^{3-/4-}$. The adsorption takes place when the electrode immersed in the solution. When an electrochemical redox reaction occurs, the inner-sphere electron transfer reaction takes place on the graphene surface, and changes the carrier density of the graphene. We had tried other outer-sphere redox molecules that will not adsorb on the electrode surface, for example $\text{Ru}(\text{NH}_3)_6^{2+/3+}$, and no redox current can be observed by GEM.